# Hypoglycaemic and Antioxidant Properties of *Acrocomia aculeata* (Jacq.) Lodd Ex Mart. Extract Are Associated with Better Vascular Function of Type 2 Diabetic Rats

**DOI:** 10.3390/nu13082856

**Published:** 2021-08-20

**Authors:** Tamaeh Monteiro-Alfredo, Sara Oliveira, Andreia Amaro, Daniela Rosendo-Silva, Katia Antunes, Ana Salomé Pires, Ricardo Teixo, Ana Margarida Abrantes, Maria Filomena Botelho, Miguel Castelo-Branco, Raquel Seiça, Sónia Silva, Kely de Picoli Souza, Paulo Matafome

**Affiliations:** 1Institute of Physiology, Faculty of Medicine, University of Coimbra, 3000-548 Coimbra, Portugal; tamaehamonteiro@hotmail.com (T.M.-A.); sara.1996.oliveira@hotmail.com (S.O.); andreia.amaro15@hotmail.com (A.A.); daniela.silva26@hotmail.com (D.R.-S.); rseica@fmed.uc.pt (R.S.); 2Coimbra Institute of Clinical and Biomedical Research (iCBR), Faculty of Medicine, Center for Innovative Biomedicine and Biotechnology (CIBB), University of Coimbra, 3000-548 Coimbra, Portugal; a.salome.pires@gmail.com (A.S.P.); ricardo.teixo@gmail.com (R.T.); margaridaabrantes@gmail.com (A.M.A.); mfbotelho@fmed.uc.pt (M.F.B.); sonias@ci.uc.pt (S.S.); 3Clinical Academic Center of Coimbra, 3000-548 Coimbra, Portugal; 4Research Group of Biotechnology and Bioprospecting Applied to Metabolism (GEBBAM), Federal University of Grande Dourados, Dourados 79825-070, MS, Brazil; avilacatia@hotmail.com (K.A.); kelypicoli@gmail.com (K.d.P.S.); 5Institute of Biophysics, Faculty of Medicine, University of Coimbra, 3000-548 Coimbra, Portugal; 6Visual Neuroscience Laboratory, Institute for Biomedical Imaging and Life Sciences (IBILI), Faculty of Medicine, University of Coimbra, 3000-548 Coimbra, Portugal; mcbranco@fmed.uc.pt; 7Centre for Neuroscience and Cell Biology (CNC), IBILI, University of Coimbra, 3000-548 Coimbra, Portugal; 8Institute for Nuclear Sciences Applied to Health (ICNAS), University of Coimbra, 3000-548 Coimbra, Portugal; 9Laboratório de Bioestatística Médica, Faculty of Medicine, University of Coimbra, 3000-548 Coimbra, Portugal; 10Institute of Pharmacology and Experimental Therapeutics, Faculty of Medicine, University of Coimbra, 3000-548 Coimbra, Portugal; 11Instituto Politécnico de Coimbra, Coimbra Health School (ESTeSC), Department of Complementary Sciences, 3000-548 Coimbra, Portugal

**Keywords:** diabetes, *macaúba*, *bocaiúva*, vascular function, polyphenols

## Abstract

Oxidative stress is involved in the metabolic dysregulation of type 2 diabetes (DM2). *Acrocomia aculeata* (Aa) fruit pulp has been described for the treatment of several diseases, and recently we have proved that its leaves have phenolic compounds with a marked antioxidant effect. We aimed to assess whether they can improve metabolic, redox and vascular functions in DM2. Control Wistar (W-Ctrl) and non-obese type 2 diabetic Goto–Kakizaki (GK-Ctrl) rats were treated for 30 days with 200 mg.kg^−1^ aqueous extract of Aa (EA-Aa) (Wistar, W-EA-Aa/GK, GK-EA-Aa). EA-Aa was able to reduce fasting glycaemia and triglycerides of GK-EA-Aa by improving proteins related to glucose and lipid metabolism, such as GLUT-4, PPARγ, AMPK, and IR, when compared to GK-Ctrl. It also improved viability of 3T3-L1 pre-adipocytes exposed by H_2_O_2._ EA-Aa also increased the levels of catalase in the aorta and kidney, reduced oxidative stress and increased relaxation of the aorta in GK-treated rats in relation to GK-Ctrl, in addition to the protective effect against oxidative stress in HMVec-D cells. We proved the direct antioxidant potential of the chemical compounds of EA-Aa, the increase in antioxidant defences in a tissue-specific manner and hypoglycaemic properties, improving vascular function in type 2 diabetes. EA-Aa and its constituents may have a therapeutic potential for the treatment of DM2 complications.

## 1. Introduction

Diabetes mellitus (DM) is a chronic disease characterized by hyperglycaemia, resulting from a deficiency in insulin production, desensitization of its action, or both [1]. According to the International Diabetes Federation, diabetes is one of the diseases with the highest incidence in the 21st century, having increased three-fold in the last two decades, and being estimated to affect 463 million individuals in 2019 [2]. As a consequence of hyperglycaemia, both DM1 and DM2 commonly have associated complications, which have significant morbidity and mortality and a considerable economic impact. These complications can be either micro (neuro, nephro, cardio and retinopathy) or macro vascular (stroke and cardiovascular diseases) [3,4]. One of the main factors for its cause and progression is oxidative stress, which is involved in the pathogenesis of complications through the overproduction of reactive oxygen species (ROS) and reactive nitrogen species (RNS), further impairing the redox balance [5]. The adoption of a better lifestyle (balanced diet and exercise) may delay the development of diabetes and its complications. On the other hand, several therapeutic options are available for DM2, most of them targeted for the reduction of the glycaemia [meglitinides, biguanides, sulfonylureas (SUs), thiazolidinedione (TZD), dipeptidyl peptidase 4 (DPP-4) inhibitors, GLP-1 receptor agonists, sodium glucose cotransporters inhibitors (SGLT2) and insulin] [6]. However, a considerable number of patients do not adhere to the treatment with allopathic medicine due to side effects [4], which affect mainly liver and kidney [7]. In this perspective, a considerable amount of research is focused on developing therapeutic alternatives, which could be both inexpensive and effective with fewer side effects [4].

Brazil has the greatest biodiversity in the world, with about 20% of the species distributed in its biomes. In particular, the Brazilian Cerrado, which occupies 22% of the national territory [8], has one of the world’s richest flora, where 35% of the species are endemic [9]. Among these species, many of them are considered medicinal due to their chemical composition [10]. *Acrocomia aculeata* (Jacq.) Lodd. ex Mart., commonly known as *bocaiúva* or *macaúba*, is a palm native from Cerrado with therapeutic (production of remedies based on ethnopharmacological knowledge—as antidiabetic, antioxidant, analgesic etc.) and economic importance (for cooking, biodiesel, and the cosmetic industry) [11]. Its carotenoid-rich fruit pulp was suggested to have beneficial effects on the treatment of respiratory diseases, as analgesic and laxative [12] and also in decreasing serum cholesterol and glucose levels [13]. Recently, our group proved the antioxidant potential of its leaves and the relevant chemical composition, mostly of vanillic, caffeic, ferulic and gallic acid, rutin and quercetin [11]. In addition to the new findings described by us about the potential of its leaves, it is only known that they are used for bovine nutritional supplementation and in the preparation of teas for human consumption.

Therefore, our goal in this study was to assess whether the antioxidant potential of *A. aculeata* leaves can restore redox balance and improve metabolic and vascular function of type 2 diabetic rats, as well as to disclose the underlying mechanisms in tissues involved in glucose metabolism and its vascular complications. Our results show the antioxidant and hypoglycaemic potential of EA-Aa, observed through the tissue-dependent upregulation of pathways involved in antioxidant defences and glucose and lipid metabolism. Such effects were associated with the improvement of aortic relaxation and redox state.

## 2. Materials and Methods

### 2.1. Chemicals and Antibodies

Salts and organic solvents used in this study were all purchased from Lonza, Sigma-Aldrich/Merck, Alfa-Aesar, Fischer Scientifics and Panreac. Antibodies used were targeted to Catalase, Glo-1, GLUT2 (ab76110, ab96032, ab54460 Abcam, Cambridge, UK), GLUT4, PPARgamma, Insulin Receptor, AMPK, phospho-AMPK-Thr-172, Sirt1, phospho-Sirt1-Ser47 (#2213S, #2443S, #3025S, #2532S, #2535S, #9475S, #2314S, Cell Signaling Technology, Danvers, MA, USA) NRF2 (sc-518036, Santa Cruz Biotechnology, Dallas, TX, USA) phospho-NRF2 (Ser40) (PA5-67520, Invitrogen, Waltham, MA, USA). Calnexin and GAPDH (AB0037, AB0049-20, Sicgen, Carcavelos, Portugal) were used as loading control.

### 2.2. Botanical Material and Isolation of Extract

Fresh *A. aculeata* leaves were collected as before [11] in the region of Grande Dourados, Macaúba district, state of Mato Grosso do Sul (MS) (22°0702.4 S 54°2836.3 W), with the permission of the Brazilian Biodiversity Authorization and Information System (*Sistema de Autorização e Informação sobre Biodiversidade*, SISBIO; no. 50589). A plant taxonomist identified the species, and a specimen was deposited in the herbarium (DDMS-UFGD) of the Federal University of Grande Dourados, Dourados (MS), Brazil, registration number—5103. The aqueous extract was prepared as previously described [11].

### 2.3. Cell Culture and Viability Assays

Mouse (*Mus musculus*) preadipocyte—3T3-L1 cells (cultured with Dulbecco’s Modified Eagle’s Medium—DMEM supplemented with 10% FBS and 1% penicillin/streptomycin) [14]; and human dermal microvascular endothelial Cells (HMVec-D, cultured with EGMTM-2, Endothelial Cell Growth Medium-2, BulletKitTM) [15,16], maintained at 37 °C and 5% CO_2_ were used in the assays.

To evaluate cell viability, 1 × 10^5^ HMVec-D cells and 3 × 10^4^ 3T3-L1 cells were seeded in 96-well microplates. After 24 h, cells were incubated with different concentrations (31,25–500 µg.mL^−1^) of EA-Aa for 24 h. After this period, cell viability was determined through the Alamar Blue assay. Absorbance was measured at 570 nm and 600 nm in a BioTek microplate reader (BioTek, Instruments, Inc., Winooski, VT, USA) and used to calculate cell viability, according to Equation (1) [11].
(1)$$\mathrm{Cell}\;\mathrm{viability}=\left(\frac{\left({\mathrm{Abs}}_{570}{-\mathrm{Abs}}_{600}\right)\mathrm{of}\;\mathrm{treated}\;\mathrm{cells}}{\left({\mathrm{Abs}}_{570}{-\mathrm{Abs}}_{600}\right)\mathrm{of}\;\mathrm{control}\;\mathrm{cells}}\right)\;\times \;100$$

To evaluate the antioxidant potential of EA-Aa, both cell lines 3T3-L1 and HMVec-D cells were treated with H_2_O_2_, the oxidative stress inductor. After 80% of confluence, cells were firstly incubated with the extract for 30 min followed by H_2_O_2_ (IC_50_ 0.125 mM in 3T3-L1 cells and 0.25 mM in HMVec-D cells) for 2 h. Equation (1) was used to calculate the protective effect of EA-Aa in cell viability. Dependence of EA-Aa effects on NRF2 pathway was evaluated through the incubation of HMVec-D cells with the NRF2 inhibitor ML385 (20 µM).

### 2.4. Animal Maintenance and Treatment

The study was performed according to good practices of animal handling, with the approval of the Institutional Animal Care and Use Committee (ORBEA 13/2018) and the procedures performed by licensed users by the Federation of Laboratory Animal Science Associations (FELASA), conformed to the guidelines from Directive 2010/63/EU of the European Parliament for the Protection of Animals Used for Science Purpose. Male 12-week-old Wistar and non-obese type 2 diabetic Goto–Kakizaki (GK) rats from our breeding colonies (Faculty of Medicine, University of Coimbra), were randomly divided in 4 groups (*n* = 5–7), as presented in Figure 1A, which were: Wistar control (W); Wistar treated with EA-Aa (W-EA-Aa); GK control (GK) and GK treated with EA-Aa (GK-EA-Aa). Animals were kept under standard conditions—2 animals per cage, with temperature at 22–24 °C, and 50–60% humidity, and standard light cycle (12 h light/12 h darkness), with water and food (standard diet A03, SAFE, France) ad libitum [17]. EA-Aa (200 mg.kg^−1^) was added in the daily water of the animals 28 days, which received the treatment during the night and normal water during the day. The weekly average of the rats’ weight was used to determine the daily dose of EA-Aa per cage.

#### 2.4.1. In Vivo Procedures and Sample Collection

Body weight, fasting glycaemia, water and food intake were evaluated weekly (calculated as the mean daily consumption per rat), and an insulin tolerance test (ITT) was performed at the beginning and at the end of the treatment. For the IIT, 250 mU.kg^−1^ insulin (Humulin, 1000 UI.mL^−1^ Lilly, Lisboa, Portugal) was injected (i.p.) after 6 h fasting, followed by glycaemia measurement in the tail vein with a glucometer (Precision Xtra Meter, Abbott Diabetes Care, Amadora, Portugal) and test strips (Abbott Diabetes Care, Portugal) at time 0, 15, 30, 60 and 120 min. Response to insulin was expressed by area under the curve (AUC) [17]. Serum triglycerides were measured in the same day before insulin administration. At the end of the treatment, animals were anesthetized (i.p.) with 2:1 (*v*/*v*) 50 mg.kg^−1^ ketamine (100 mg.mL^−1^)/2.5% chlorpromazine (5 mg.mL^−1^) and samples of blood were collected by cardiac puncture followed by cervical dislocation. Epididymal adipose tissue (EAT), liver, kidney, heart and aorta were collected, blood samples were centrifuged (2200× *g*, 4 °C, 15′) and serum and plasma were aliquoted and stored at −80 °C for further analysis.

#### 2.4.2. Studies of Isometric Tension of Aorta

Aorta rings were mounted on stainless steel hooks under 19.6 mN basal tension in organ baths filled with aerated (95% O_2_, 5% CO_2_) Krebs–Henseleit solution (37 °C, pH 7.4) (NaCl 118.67 mmol/L; KCl 5.36 mmol/L; CaCl_2_ 1.90 mmol/L; MgSO_4_ 0.57 mmol/L; NaHCO_3_ 25.00 mmol/L; KH_2_PO_4_.H_2_O 0.90 mmol/L; glucose 11.1 mmol/L). After an equilibration period of 60 min, aortic rings were precontracted with 10 µM of noradrenaline and cumulative isometric concentration-response curves were performed in response to acetylcholine (ACh) (0.01 to 90 µM) in the presence and absence of 100 µM ascorbic acid. Cumulative curves were recorded with Letica Scientific Instruments isometric transducers connected to a four-channel polygraph (Polygraph 4006, Letica Scientific Instruments, Barcelona, Spain).

### 2.5. Biochemical Analyses

Plasma insulin and free fatty acids (FFA) tests were determined through the Rat Insulin ELISA Kit (Mercodia, Uppsala, Sweden) and FFA Assay Kit (ZenBio, Research Triangle Park, NC, USA), according to the manufacturers’ instructions. Heart 8-Isoprostane levels were determined using an ELISA Kit according to the manufacturer’s instruction (Cayman Chemical, Ann Arbor, MI, USA).

### 2.6. Fluorescence Immunocytochemistry and Immunohistochemistry

The evaluation of the antioxidant potential of EA-Aa was carried out in HMVec-D cells challenged with H_2_O_2_ (same as the antioxidant assay described before), and in cryopreserved histological slices (4 µm) of liver, kidney, and aorta of the animal models. Oxidative stress probes, 2,7-dichlorodihydrofluorescein diacetate (H_2_DCFDA) and dihydroethidium (DHE) were used and DAPI was used to stain the nucleus. In hydrated sections, probes were incubated for 30 min and the slices were mounted with mounting medium (Glycergel, DAKO, Carpinteria, CA, USA). Images were immediately obtained with a fluorescence microscope (Zeiss Axio Observer Z1) with an incorporated camera (Zeiss, Jena, Germany), detected with 504 nm of excitation and 525 nm of emission for DCF, 587 nm of excitation and 610 nm of emission for DHE, and 353 nm of excitation and 465 nm of emission for DAPI. The same settings were kept constant for all analysis and the entire image was used for quantification, which was performed with ImageJ software.

### 2.7. Western Blot

The Western blot analysis were performed in both cells (3T3-L1 cells treated with EA-Aa 31.25–500 µg.mL^−1^ for 24 h) and organs (EAT, heart, kidney, liver and aorta). Cells and organ samples were washed with PBS and disrupted in lysis buffer (0.25 M Tris-HCl, 125 mM NaCl, 1% Triton-X-100, 0.5% SDS, 1 mM EDTA, 1 mM EGTA, 20 mM NaF, 2 mM Na_3_VO_4_, 10 mM β-glycerophosphate, 2.5 mM sodium pyrophosphate, 10 mM PMSF, 40 µL of protease inhibitor) using the TissueLyser systems (Quiagen, Germany). The BCA Protein Assay Kit was carried out on the supernatant of the centrifugation of samples (14.000 rpm for 20 min at 4 °C), followed by the addition of Laemmli buffer (62.5 mM Tris-HCl, 10% glycerol, 2% SDS, 5% β-mercaptoethanol, 0.01% bromophenol blue) [18]. Samples (20 μg) were loaded into SDS-PAGE and electroblotted onto PVDF membrane (Advansta, San Jose, CA, USA). Membranes were blocked with TBS-T 0.01% and BSA 5%, then incubated with the primary (overnight, 4 °C) and secondary antibodies (2 h, room temperature), following the dilutions suggested by the manufacturers. Immunoblots were detected with ECL substrate and the Versadoc system (Biorad, Hercules, CA, USA).

### 2.8. Statistical Analysis

Data were expressed as the mean ± standard error of the mean (SEM) and compared by analysis of variance using the Kruskal–Wallis test or ANOVA followed by the Tukey post hoc test, according to normality evaluation. Student’s *t*-test was used to determine the differences between two groups. Values of *p* < 0.05 were considered significant. Statistical tests were performed with GraphPad Prism 5.0 and IBM SPSS Statistics Software.

## 3. Results

### 3.1. EA-Aa Improves the Metabolic Profile of Diabetic Rats

After the 30-day treatment (presented in the experimental design, Figure 1A), animals did not show any alterations in body weight and food/caloric intake among groups (Figure 1B–D). However, the treatment with EA-Aa decreased the fasting hyperglycaemia of diabetic rats (GK-EA-Aa) by 30–40% along the treatment in comparison to both controls at all timepoints (Figure 1E). Similar data were observed for fasting glycaemia, where the initial difference between GK-Ctrl and W-Ctrl groups becomes non-significant after the treatment in GK-EA-Aa (Figure 1F), and for the AUC of glycaemia along the 4-week treatment period, which was also significantly reduced in GK-EA-Aa when compared to GK-Ctrl (Figure 1G). No significant effects of EA-Aa were observed in the AUC of the insulin tolerance test (Figure 1H). On the other hand, a significant increase of plasma insulin levels was observed in GK-EA-Aa in relation to W-Ctrl rats (Figure 1I). Regarding lipid metabolism, GK-Ctrl presented higher levels of triglycerides, which were reduced in the GK-EA-Aa group (Figure 1J), whereas the values of FFA showed no significant difference (Figure 1K).

The epididymal adipose tissue and the liver were analysed to understand the mechanisms of EA-Aa-induced metabolic improvement. The haematoxylin-eosin staining presented no morphological alterations and no significant weight changes were observed after EA-Aa treatment in the EAT and liver of Wistar and diabetic rats, as shown in Figure 2A,B,G–H. In addition, GLUT4 and PPARγ levels in EAT were increased in the diabetic rats treated with EA-Aa (Figure 2C,F). No changes were observed for the insulin receptor levels nor AMPK (Figure 2D,E), although a trend to higher AMPK levels was observed in diabetic rats after EA-Aa treatment. In the liver, EA-Aa partially restored the levels of the insulin receptor in GK rats (64% *vs* 38% of %W-Ctrl/Calnexin). No significant differences were observed in hepatic GLUT2 and AMPK levels (Figure 2I,K).

### 3.2. EA-Aa Has Tissue-Specific Protective Antioxidant Effects

To determine the antioxidant potential of EA-Aa in vivo, Sirt1, NRF2, catalase, SOD-1 and GLO-1 levels were evaluated in EAT, liver, heart and kidney of control and diabetic rats. Markers of oxidative stress were evaluated and, given the metabolic alterations in adipose tissue caused by EA-Aa, the protective antioxidant effects were evaluated in a 3T3-L1 preadipocyte cell line. In the adipose tissue (Figure 3A,C,D) and liver (Figure 3I–L), no significant differences were observed between control and diabetic rats in Sirt1, catalase, GLO-1 and SOD-1 levels, whereas a partially restoration of NRF2 levels was observed in EAT of GK-EA-Aa group (Figure 3B). In the liver, the histological staining of a superoxide anion probe (DHE) did not show differences between the experimental groups as well (Figure 3H). Given that the antioxidant potential of EA-Aa was already confirmed in Cos-7 cells [11], we evaluated such EA-Aa effect in 3T3-L1 cells challenged with an oxidant stimulus. After confirming the absence of EA-Aa-induced toxicity (Figure 3E), we incubated cells with H_2_O_2_ (IC_50_–0.125 mM), which confirmed the protective antioxidant effect of the extract (500 μg./mL^−1^) against H_2_O_2_-induced oxidative stress (Figure 3F). No activation of Sirt1 and NRF2 pathways was observed suggesting that the effect is at least partially independent of such pathways (representative Western Blots at Figure 3G).

Heart and kidney were analysed given their high susceptibility to hyperglycaemia-driven complications. In both organs, no significant alterations were observed in Sirt1, NRF2, GLO-1 and SOD-1 levels (Figure 4A–D,I). Nevertheless, EA-Aa treatment resulted in a significant increase of kidney catalase levels of both normal and diabetic rats (Figure 4H). No significant alterations were observed in the heart levels of the lipid peroxidation marker 8-Isoprostane, as well as in kidney histological analysis of morphology (Figure 4E,G) and superoxide anion (DHE, Figure 4F). A trend to reduced DHE staining in the glomerulus was observed in some kidney regions after EA-Aa treatment, although quantification did not show significant differences (data not shown).

### 3.3. EA-Aa Improves Diabetic Endothelial Dysfunction Reducing Vascular Oxidative Stress

Given that endothelial dysfunction is one of the major complications of hyperglycaemia, aorta relaxation was evaluated after EA-Aa treatment in Wistar and diabetic rats. Vascular relaxation was evaluated in NA-precontracted aorta rings in response to Ach in the presence or absence of ascorbic acid [19]. At 12-week-old, Wistar and GK rats had similar ACh-dependent relaxation (Figure 5A). In the rats treated with EA-Aa, a slight effect on aorta relaxation was observed, namely a 15% increment between W-EA-Aa and W-Ctrl (Figure 5B) and 10% between GK- EA-Aa and GK-Ctrl (Figure 5C), although statistical significance was not reached. Pre-incubation of the rings with L-NAME (NOS inhibitor) practically abolished the relaxation of the aortic rings (data not shown), showing endothelial dependence. When the aorta was incubated with ascorbic acid, the maximum endothelial-dependent relaxation mediated by NA-precontracted rings in response to Ach reached 40% more in W-Ctrl and almost 74% more in W-EA-Aa (Figure 5D–E). Interestingly, no response to ascorbic acid was obtained in GK-Ctrl (Figure 5F), but it was restored after the treatment with the extract, which promoted a 30% increment of relaxation in GK-EA-Aa (Figure 5G). Such results are supported by the histological fluorescence staining of both oxidative stress probes, DHE and DCF, where 34% and 67% respectively lower signal intensity was observed in GK-EA-Aa in relation to the GK-Ctrl group (Figure 6A–D). The treatment also promoted an increase of catalase levels in the aorta of treated groups from both strains, mainly in diabetic rats (Figure 6E).

To further assess the antioxidant effect of EA-Aa in the vessel wall, we used a microvascular endothelial cell line, HMVec-D, in which the absence of toxicity of the EA-Aa can be observed in Figure 6F. In cells with H_2_O_2_-induced oxidative stress, the EA-Aa had a protective effect in the concentrations of 125, 250 and 500 μg.mL^−1^, which is represented by a ~22% improvement of cell viability (Figure 6G). Cells incubated with NRF2 inhibitor, ML385, presented a decrease in cell viability of almost 52% when induced with H_2_O_2_, which has been improved by EA-Aa treatment in 10% to 20% (according to the increase in concentration). Thus, the protective effects of the EA-Aa were only attenuated in the presence of ML385, suggesting a protection partially mediated by NRF2 pathway, but mostly resulting from the direct antioxidant properties of the extract (Figure 6H).

## 4. Discussion

Medicinal plants may be a therapeutic alternative since they have long been applied in the treatment of several diseases, showing important scientifically proven results in improving health condition in several diseases with low side effects and costs [20]. Considering the large consumption of plants and people’s adherence to their use as a therapeutic strategy, the relationship between oxidative stress and the development of DM2 and its complications [5,19,21,22], and the relevant antioxidant potential that EA-Aa previously demonstrated [11], we aimed to analyse its potential as a therapeutic strategy in the treatment of DM2 and its complications. The present study provides evidence for the in vitro and in vivo antioxidant potential of EA-Aa, which is associated to the improvement of metabolic profile and vascular function of diabetic rats.

The aqueous extract of *A. aculeata* (EA-Aa) is mainly composed by phenolic compounds and flavonoids, which are probably involved in the therapeutic effects of the extract. Similar data have already been described for the fruit pulp [23] and kernel of *A. aculeata* [24,25,26]. Plants of the same family, *Arecaceae,* have similar composition as EA-Aa, namely caffeic acid, rutin and quercetin [27,28,29,30].

The different therapeutic effects reported here, are in line with previous demonstrations regarding the mechanisms of action of some of the individual compounds of the extract. The more notable effect in the phenotype of the diabetic animals was the reduction of hyperglycaemia, which could be associated to the ferulic (4-hydroxy-3-methoxycinnamic acid), caffeic (3,4-dihydroxycinnamic acid) and gallic (3,4,5-trihydroxybenzoic acid) acids [31]. Ferulic acid seems to reduce glycaemia through the suppression of the activity of the enzyme α-glucosidase and stimulation of insulin secretion [32]. Caffeic acid was associated to higher insulin levels and glucose uptake through AMPK pathway [31,33,34]. Moreover, gallic acid was observed to ameliorate hyperglycaemia and HOMA-IR index [35] and to induce GLUT4 translocation to the plasma membrane [36]. Regarding the flavonoids found in EA-Aa, quercetin (3,5,7-trihydroxy-2-(3,4-dihydroxyphenyl)-4Hchromen-4-one) was also associated to hypoglycaemic mechanisms [32] such as higher levels of insulin receptor (IR) and insulin receptor substrate (IRS), besides GLUTs and the inhibition of α-glucosidase activity, which are associated to the improvement of insulin resistance [37]. Rutin has been described to increase PPARγ expression and glucose uptake, in addition to also being associated with better insulin sensitivity [38] and the inhibition of α-glucosidase and α-amylase [39]. Similar effects were noted, in particular in the group GK-EA-Aa, namely the decrease in glycemia and the increased levels of AMPK in EAT. Also considering the absence of effects in the ITT (acute insulin action) and the reduction of fasting triglycerides levels, we can consider the mechanism of action more at the level of energy balance stabilization and reduction of oxidative stress, which ends up in a better long-term metabolic function at a systemic level [40,41,42,43]. This can be observed in GK-Ctrl, which presented fasting hyperglycaemia and hypertriglyceridaemia. Thus, treatment with EA-Aa reduced triglyceridaemia at baseline levels, which is probably associated with the improvement of AMPK, a known energy sensor and metabolic regulator [44].

Besides the improvement in the metabolic profile of the animals, treatment with EA-Aa also improved the hyperglycaemia-driven endothelial dysfunction. This beneficial effect was previously attributed to the flavonoids rutin [45] and quercetin [46] and the phenolic compounds [47], ferulic [48,49], vanillic, caffeic [46] and gallic acids [50], which may be acting in synergy. An imbalance in the redox state is developed as a consequence of hyperglycaemia, exacerbating ROS production. This effect, together with the reduction of antioxidant defence systems, decreases nitric oxide bioavailability and leads to endothelial dysfunction and vascular damage [19,51]. Thus, considering the reduction in fluorescence probes DHE and DCF, the increased levels of catalase in aorta and the protective effect in HMVec-D cells suggest that the effect of EA-Aa occurs by a stabilization of the redox balance in the organ. It is questionable, however, whether such effects are a consequence of improved glucose metabolism or modulation of antioxidant systems. In fact, an activation of Sirt-1–NRF2 was not detectable in vivo and only a partial dependence was observed in vitro. Moreover, both the reduction of glycaemia and the upregulation of catalase were observed in diabetic rats, suggesting that EA-Aa may have direct antioxidant effects (as previously demonstrated here), but also modulate such pathways in vivo, which lead us to believe in a long-term effect by EA-Aa.

Several studies have revealed the pharmacological potential of the different phytochemicals that are metabolized by gut microbiota, which present distinct therapeutic effects than the crude extract [52]. Compounds presented in EA-Aa such as gallic, caffeic, ferulic and vanillic acids, rutin and quercetin, all have metabolites produced after in vivo metabolism. Some of them derived from gallic acid include pyrogallol-1-O-glucuronide, 4-OMeGA, 4-OMeGA-3-O-sulfate, pyrogallol-O-sulfate, deoxypyrogallol-O-sulfate, and O-methylpyrogallol-O-sulfate [53]. Ferulic acid is a metabolite of caffeic acid found in the gut lumen [54], which also forms ferulic acid-4-O-glucuronide, ferulic acid-4-O-sulfate, feruloylglycine, and dihydroferulic acid [55], besides being converted into vanillic acid [56]. Moreover, the deglycosylation that happens in the gut lumen forms the quercetin metabolites, quercetin sulfate and quercetin glucoronides [52,57]. Rutin originates the metabolites 3,4-dihydroxyphenylacetic acid, 3,4-dihydroxyphenylacetic acid (DHPAA), 3,4-dihydroxytoluene (DHT), 3-hydroxyphenylacetic acid (HPAA), and 4-hydroxy-3-methoxyphenylacetic acid (homovanillic acid, HVA) [58].

Although the individual contribution of such compounds/metabolites and secondary metabolites is unknown, our results support the effects of EA-Aa at the metabolic level, in improving the glycaemic and lipidic profile and vascular function. These results are probably associated to the mechanistic effect of the compounds present in EA-Aa that may act synergistically, acting to re-establish and maintain the redox balance and consequently prevent complications associated with DM2. In this way, our results support new studies enabling us to understand the mechanisms of phenolic compounds and flavonoids (especially those from EA-Aa) and its metabolites in preventing DM2 complications.

## 5. Conclusions

Taken together, our results provide evidence for the potential of EA-Aa in improving metabolic pathways in EAT and liver and reducing fasting glycaemia and triglyceride levels. In addition, the treatment with EA-Aa increased vascular redox condition and function, through direct antioxidant properties and modulation of antioxidant systems. Such a reduction of glycaemia and improvement of redox state was associated with improved vascular relaxation in response to acetylcholine, especially in the presence of ascorbic acid. The results obtained in this study suggest that although individual compounds may have a therapeutic role in diabetic complication, their natural combination in plant extracts may also exert beneficial mechanisms. Moreover, the therapeutic effects found here may be distinct if such compounds are administered through non-oral routes and their gut metabolization should be understood in the future. Therefore, the improvement of the metabolic-redox condition by EA-Aa encourages more studies using the compounds present in EA-Aa and their metabolites as a strategy for the development of treatments for the complications associated to DM2.

## Figures and Tables

**Figure 1 nutrients-13-02856-f001:**
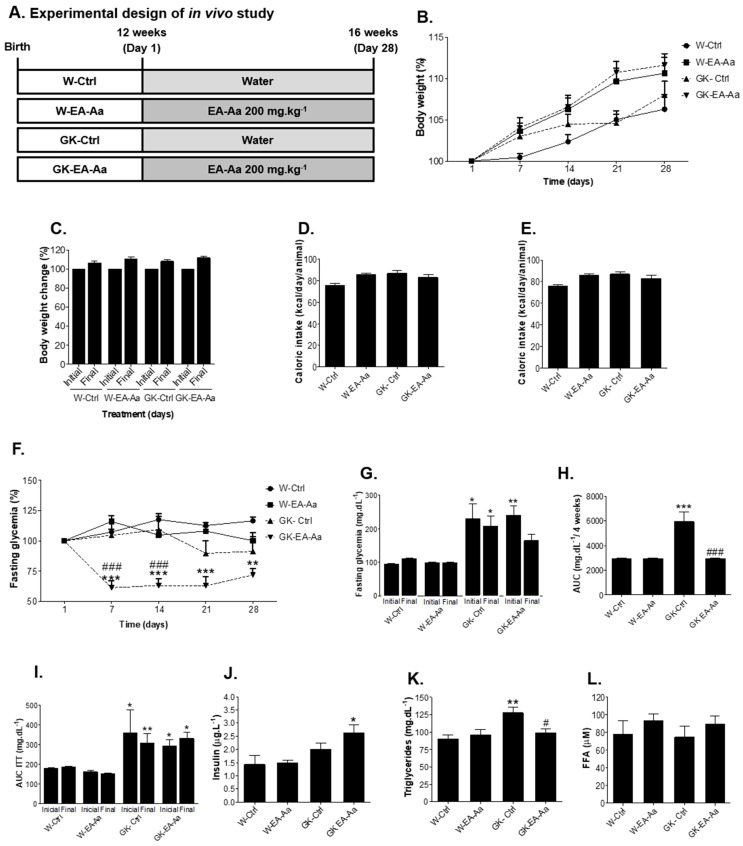
Glycaemic and lipid profile of Wistar and Goto–Kakizaki (GK) rats after 30-day treatment (*n* = 5–7). (**A**) Experimental design of in vivo study. (**B**) Body mass evolution of treated and non-treated rats represented every 7 days. (**C**) Initial and final body mass of treated animals. (**D**) Caloric intake. (**E**) Fasting blood glucose in percentage of the initial value; EA-Aa promoted a decrease in fasting glycaemia of GK-EA-Aa since the 7th day of treatment in relation to the initial glycaemia. (**F**) Fasting glycaemia; a restoration of values is found in the end of the treatment between GK rats and W-Ctrl group. (**G**) Area under the curve of glycaemia along 4-week treatment period; values are reduced by EA-Aa treatment in GK rats when comparing to the control group. (**H**) Area Under the curve of the glycaemia along all the treatment. (**I**) Area under the curve of ITT (insulin tolerance test). (**J**) Insulin; an increase in GK-EA-Aa is presented in relation to the W-Ctrl group. (**K**) Triglycerides; treatment with EA-Aa decreased plasma triglycerides levels in GK. (**L**) Free fatty acid levels. * vs. W-Ctrl at the same point; # vs. GK-Ctrl at the same point; *,# *p* < 0.05; ** *p* < 0.01; ***,### *p* < 0.001.

**Figure 2 nutrients-13-02856-f002:**
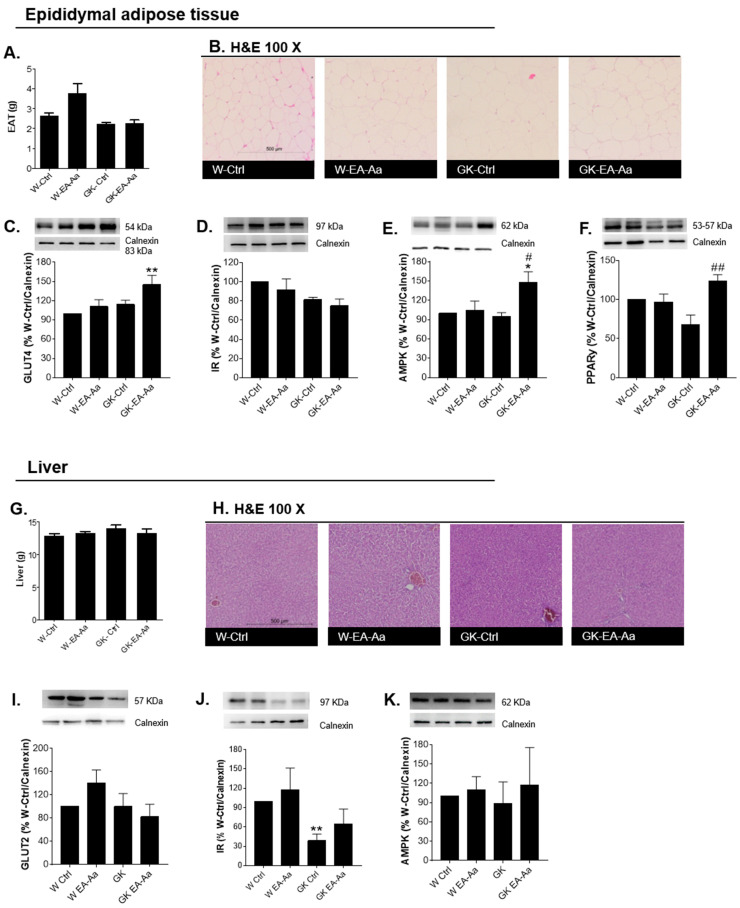
Regulation of metabolic pathways in epididymal adipose tissue (EAT) and liver. (**A**) Epididymal adipose tissue weight (*n* = 5–7). (**B**) Haematoxylin-eosin staining in EAT (*n* = 4). (**C**) GLUT4 levels in EAT; treatment with EA-Aa increased the levels of the protein in the treated diabetic group. (**D**) Insulin receptor levels in EAT. (**E**) AMPK levels in EAT; an increase in AMPK levels is presented in EA-Aa GK-treated group in relation to both controls. (**F**) PPARγ levels in EAT; GK-EA-Aa shows an increase in relation to W-Ctrl group (*n* = 5 for Western blots). (**G**) Liver weight (*n* = 5–7). (**H**) Haematoxylin-eosin staining in liver (*n* = 4). (**I**) GLUT2 levels in liver. (**J**) Insulin receptor levels in liver; treatment with EA-Aa promoted a restoration in IR levels in relation to the normal control group. (**K**) AMPK levels in liver (*n* = 5 for Western blots). * vs. W-Ctrl; # vs. GK-Ctrl; *,# *p* < 0.05; **,## *p* < 0.01.

**Figure 3 nutrients-13-02856-f003:**
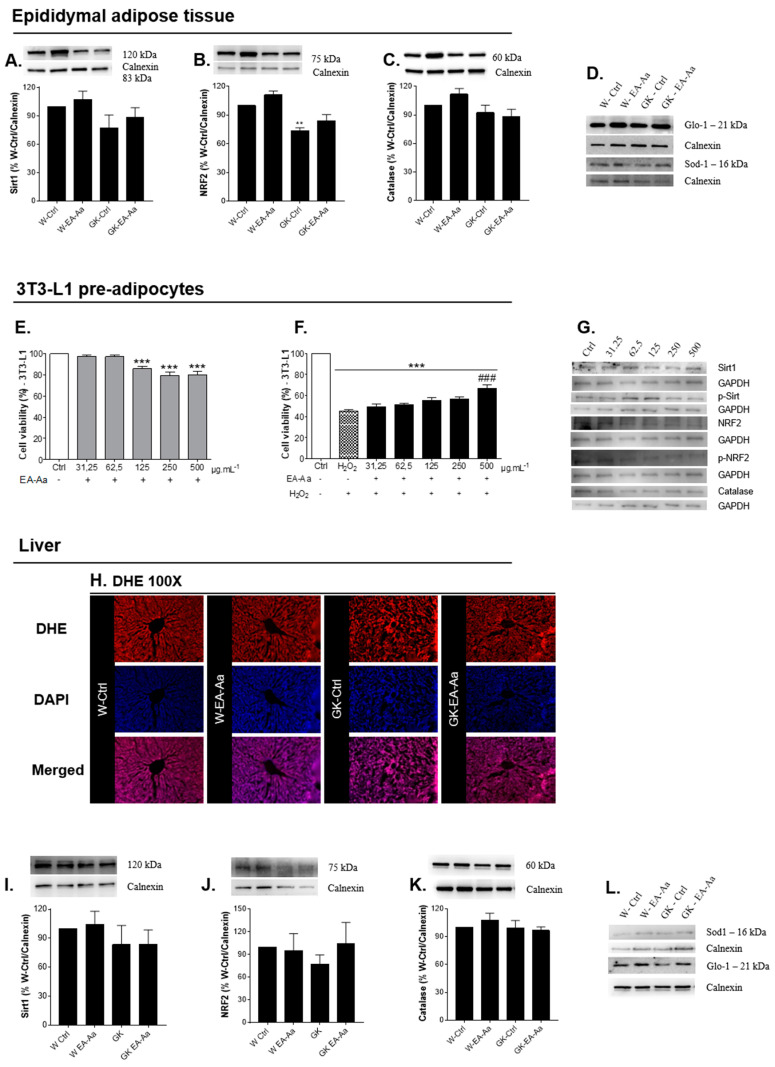
Antioxidant effects of EA-Aa in EAT, liver and in 3T3-L1 pre-adipocytes (*n* = 3 independent experiments). (**A**) Sirt1 levels in EAT. (**B**) NRF2 levels in EAT, EA-Aa promotes partial restoration of NRF2 levels in the non-obese type 2 diabetic EA-Aa-treated group. (**C**) Catalase levels in EAT. (**D**) Representative images of EAT Western blot. Cell viability of pre-adipocytes 3T3-L1: (**E**) Treatment with EA-Aa for 24 h. (**F**) Treatment with EA-Aa (previously for 30 min) and induction to oxidative stress with H_2_O_2_ (for 2 h). (**G**) Representative images of 3T3-L1 Western blot. (**H**) Dihydroethidium (DHE) staining in liver (*n* = 3). (**I**) Sirt1 levels in liver. (**J**) NRF2 levels in liver. (**K**) Catalase levels in liver (*n* = 5 for Western blots). (**L**) Representative images of liver Western blot. * vs. W-Ctrl/Ctrl; # vs. H_2_O_2_; ** *p* < 0.01; ***,### *p* < 0.001.

**Figure 4 nutrients-13-02856-f004:**
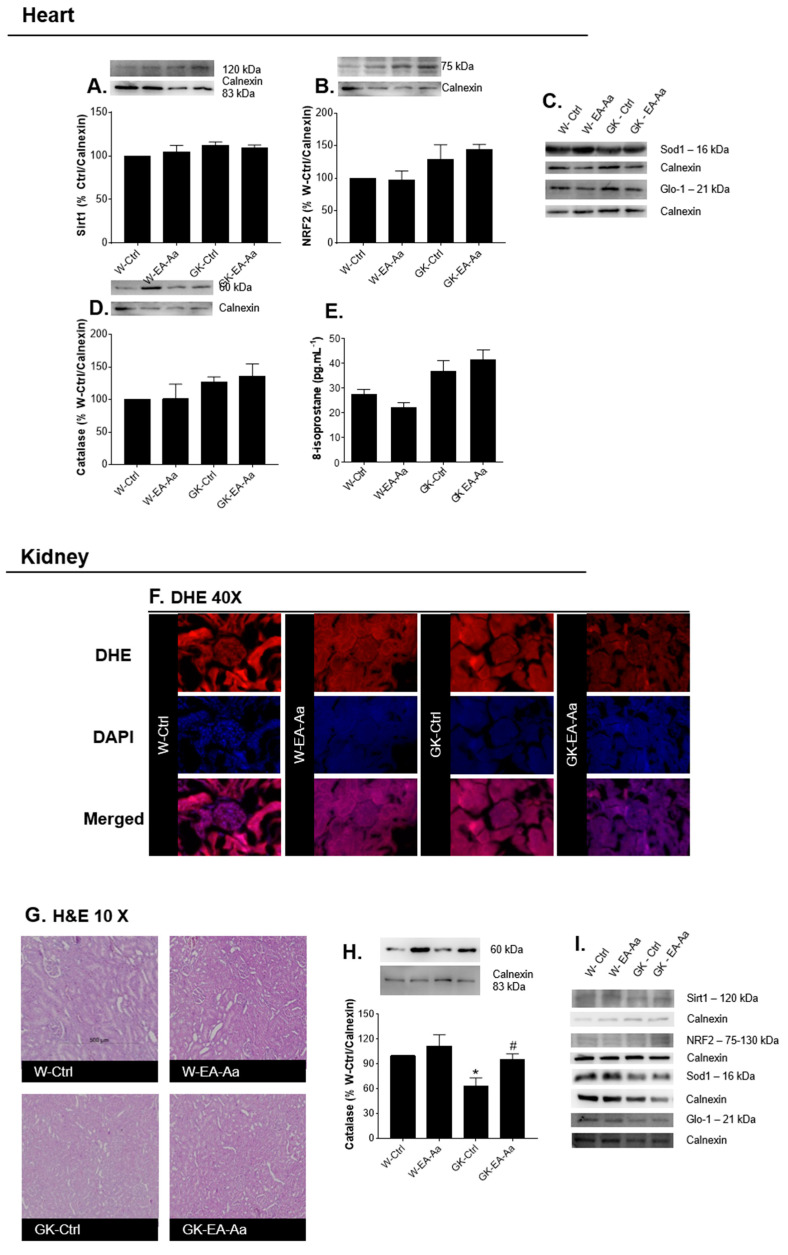
Levels of stress-related proteins in target organs of redox imbalance, heart and kidney. (**A**) Sirt1 levels in heart. (**B**) NRF2 levels in heart. (**C**) Representative images of heart Western blot. (**D**) Catalase levels in heart (*n* = 5 for Western blots). (**E**) Heart 8-isoprostane levels (*n* = 5–7). (**F**) DHE staining in kidney (*n* = 3). (**G**) Haematoxylin-eosin staining in kidney. (**H**) Catalase levels in kidney; a restoration is evident is GK-EA-Aa in relation to GK-Ctrl group. (**I**) Representative images of kidney Western blot. * vs. W-Ctrl/Ctrl; # vs. H_2_O_2_; *,# *p* < 0.05.

**Figure 5 nutrients-13-02856-f005:**
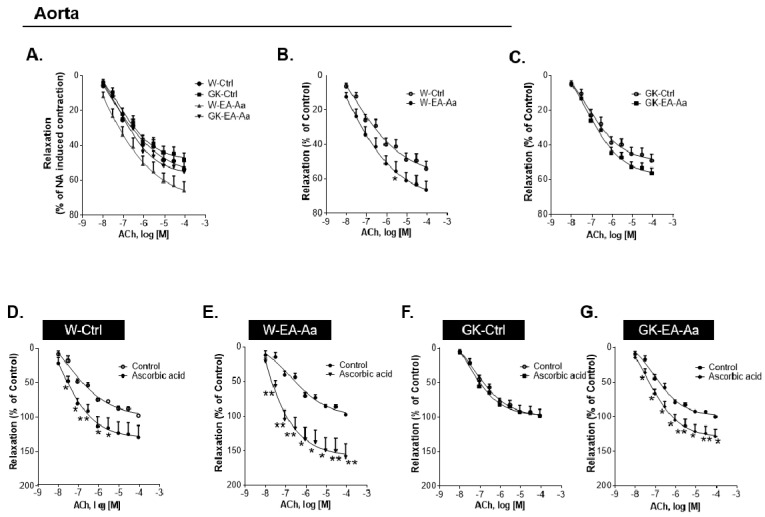
Increased isometric relaxation of aorta promoted by EA-Aa (*n* = 5-7). Vascular relaxation of NA-precontracted aorta in response to Ach: (**A**) both strains, control and treated rats. (**B**) Wistar group; treatment with EA-Aa shows a light increase in aorta relaxation compared to W-Ctrl. (**C**) GK group. Aorta pre-incubated with ascorbic acid: (**D**) W-Ctrl; an increase in aorta relaxation promoted by ascorbic acid is evident in pre-incubated W-Ctrl. (**E**) W-EA-Aa; treatment with EA-Aa increased the relaxation in normal rats in comparison to the non-pre-incubated group. (**F**) GK-Ctrl. (**G**) GK-EA-Aa; EA-Aa induces an improvement in aorta relaxation in comparison to the non-pre-incubated aorta in diabetic rats. * vs. the same point with or without ascorbic acid pre-incubation; * *p* < 0.05; ** *p* < 0.01.

**Figure 6 nutrients-13-02856-f006:**
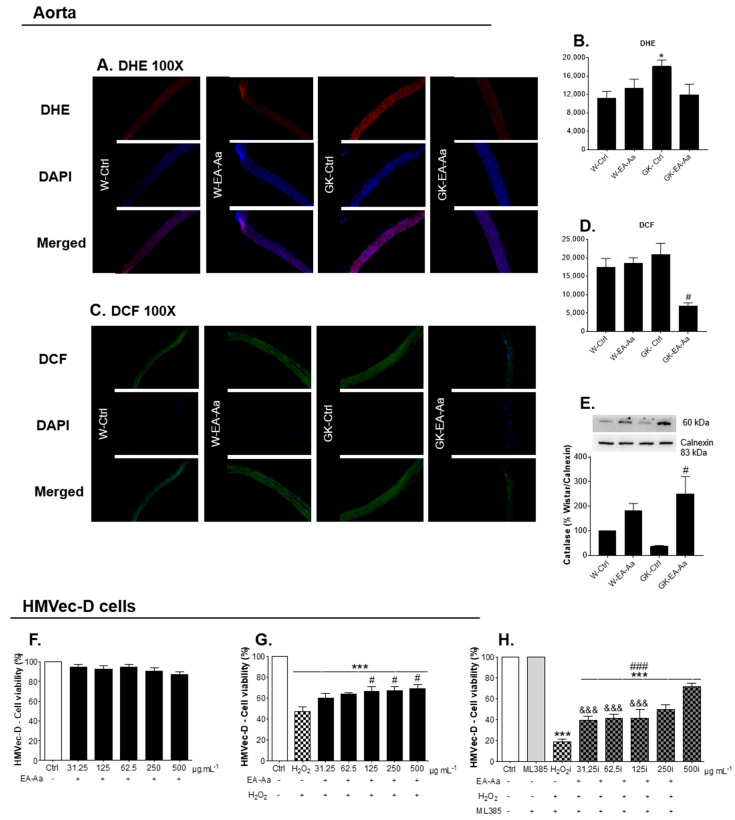
Antioxidant protection of EA-Aa at the vascular level, in aorta and in human endothelial microvascular dermal cells (HMVec-D). (**A**) DHE staining in aorta. (**B**) Fluorescence intensity of DHE; staining in aorta shows a restoration in intensity of fluorescence in GK-EA-Aa group in relation to GK-Ctrl. (**C**) DCF staining in aorta. (**D**) Fluorescence intensity of DCF; staining in aorta shows a reduction after the treatment with EA-Aa (*n* = 5). (**E**) Catalase levels in aorta; treatment with EA-Aa increases catalase levels in GK-EA-Aa (*n* = 5). Cell viability of HMVec-D cells (*n* = 3 independent experiments): (**F**) Treatment with EA-Aa for 24 h. (**G**) Previous treatment with EA-Aa (for 30 min) and induction to oxidative stress with H_2_O_2_ (for 2 h).; a protection of EA-Aa against H_2_O_2_ is evidenced since 125 µg.mL^−1^. (**H**) Pre-treatment of cells with NRF2 inhibitor, ML385, followed by incubation with EA-Aa and induction with H_2_O_2_; EA-Aa increases cell metabolic function in all concentrations and presents differences in relation to the non-inhibited group in 31.25–125 µg.mL^−1^. * vs. W-Ctrl/Ctrl; # vs. GK-Ctrl/ H_2_O_2_; & vs. same point without NRF2 inhibitor; *,# *p* < 0.05, ***,###,&&& *p* < 0.001.

## Data Availability

The datasets generated during and/or analyzed during the current study are available from the corresponding author upon reasonable request.

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
