# Peer review of "Hypoglycaemic and Antioxidant Properties of Acrocomia aculeata (Jacq.) Lodd Ex Mart. Extract Are Associated with Better Vascular Function of Type 2 Diabetic Rats"

_nutrients, 2021, doi:10.3390/nu13082856_

Round 1
Reviewer 1 Report
Overall good study and very well presented.
Need to be more specific in the methodology and also the conclusions need to be more documented. Good continuation of the articles previously written by authors
Author Response
Overall good study and very well presented.
Need to be more specific in the methodology and also the conclusions need to be more documented. Good continuation of the articles previously written by authors
We are grateful to the reviewer for the positive evaluation of our manuscript. The questions raised regarding the methodology and conclusions were addressed and the information was completed. All the manuscript was also revised.

Reviewer 2 Report
The authors describe a thorough investigation of the anti-hyperglycemic, anti-oxidant properties of Acrocomia aculeata using both in vivo and in vitro methods. The experimental designs appear sound and the results are presented appropriately. The comments below focus on the writing style, which requires considerable improvement.
Abstract: list all the rat groups and specify comparisons when describing results.
Lines 31, 83: EA-Aa - define.
Lines 45-47: sentence structure needs improvement.
Lie 67: their chemical...
68-74: break into smaller sentences.
139: with 12-weeks-old - remove "with"
148: Describe how water and food intake were calculated per rat, given that rats were housed 2/cage.
Figure 1: Change "evolution" to percent change in body weight. Indicate that values are means (SD - or whatever is correct) in all figure footnotes.
154, 221: at time...
161: describe place and length of storage of samples.
218: "...caloric intake" - please add results of food intake measures and calculate feed efficiency to test for an effect of the extract on kcal absorption/utilization.
Figures: indicate number of rats per group in the footnotes.
Figure 4: Modify the footnote to remove "improved" since some results showed no effect of the treatment.
369: "probably the responsible" - needs correction.
377, 385, 387, 434, 441, 447: associated with
401: "enhanced" is not the correct word to use
404: may be
421-423; 425-6: Sentences needs correction. This entire paragraph needs editing.
432: "support" instead of "proved."
435: maintain
Author Response
The authors describe a thorough investigation of the anti-hyperglycemic, anti-oxidant properties of Acrocomia aculeata using both in vivo and in vitro methods. The experimental designs appear sound and the results are presented appropriately.
We are grateful to the reviewer for the careful evaluation of our manuscript, as well as for the constructive comments that we used to improve the manuscript quality. Please find below a point-by-point response to all the comments, questions and suggestions raised.
The comments below focus on the writing style, which requires considerable improvement.
Abstract: list all the rat groups and specify comparisons when describing results.
Lines 31, 83: EA-Aa - define.
Lines 45-47: sentence structure needs improvement.
Lie 67: their chemical...
68-74: break into smaller sentences.
139: with 12-weeks-old - remove "with"
148: Describe how water and food intake were calculated per rat, given that rats were housed 2/cage.
Figure 1: Change "evolution" to percent change in body weight. Indicate that values are means (SD - or whatever is correct) in all figure footnotes.
154, 221: at time...
161: describe place and length of storage of samples.
218: "...caloric intake" - please add results of food intake measures and calculate feed efficiency to test for an effect of the extract on kcal absorption/utilization.
Figures: indicate number of rats per group in the footnotes.
Figure 4: Modify the footnote to remove "improved" since some results showed no effect of the treatment.
369: "probably the responsible" - needs correction.
377, 385, 387, 434, 441, 447: associated with
401: "enhanced" is not the correct word to use
404: may be
421-423; 425-6: Sentences needs correction. This entire paragraph needs editing.
432: "support" instead of "proved."
435: maintain
We thank the reviewer for the detailed review of the manuscript. All the errors identified were corrected and the manuscript was carefully revised again.
